# Origin of high thermal conductivity in disentangled ultra-high molecular weight polyethylene films: ballistic phonons within enlarged crystals

Taeyong Kim[1], Stavros X. Drakopoulos[2], Sara Ronca[2] & Austin J. Minnich [1✉]

The thermal transport properties of oriented polymers are of fundamental and practical interest. High thermal conductivities ($\gtrsim 50$ Wm$^{-1}$K$^{-1}$) have recently been reported in disentangled ultra-high molecular weight polyethylene (UHMWPE) films, considerably exceeding prior reported values for oriented films. However, conflicting explanations have been proposed for the microscopic origin of the high thermal conductivity. Here, we report a characterization of the thermal conductivity and mean free path accumulation function of disentangled UHMWPE films (draw ratio ∼200) using cryogenic steady-state thermal conductivity measurements and transient grating spectroscopy. We observe a marked dependence of the thermal conductivity on grating period over temperatures from 30–300 K. Considering this observation, cryogenic bulk thermal conductivity measurements, and analysis using an anisotropic Debye model, we conclude that longitudinal atomic vibrations with mean free paths around 400 nanometers are the primary heat carriers, and that the high thermal conductivity for draw ratio $\gtrsim 150$ arises from the enlargement of extended crystals with drawing. The mean free paths appear to remain limited by the extended crystal dimensions, suggesting that the upper limit of thermal conductivity of disentangled UHMWPE films has not yet been realized.

---

[1] Division of Engineering and Applied Science, California Institute of Technology, Pasadena, CA 91125, USA. [2] Department of Materials, Loughborough University, Loughborough LE11 3TU, UK. ✉email: aminnich@caltech.edu

Thermally conductive polymers are of interest for fundamental materials science as well as applications such as thermal management[1–6]. Although the thermal conductivity of unoriented polymers is generally $<1\,W m^{-1} K^{-1}$ (ref. [2]), early works reported orders of magnitude increase in uniaxial thermal conductivity of oriented samples, including polyethylene (PE)[7,8], polyacetylene[9], and polypropylene[10,11]. In particular, the reported thermal conductivity of oriented polyethylene ranged from $\sim 14\,W m^{-1}\,K^{-1}$ (ref. [12]) for draw ratio DR = 25 up to $\sim 40\,W m^{-1}\,K^{-1}$ for solution processed PE with a DR of 350 (ref. [13]). The enhancement was attributed to various mechanisms including increased chain alignment along the drawing direction[7,14], phonon focusing in the elastically anisotropic crystalline phase[15,16], and increased crystallinity[12,17]. Recently, thermal conductivity values around $20-30\,W m^{-1}\,K^{-1}$ and $\gtrsim 60\,W m^{-1}\,K^{-1}$ have been reported in PE microfibers[14,18] and nanofibers[19,20], respectively. In macroscopic samples, the introduction of disentangled ultra-high molecular weight polyethylene (UHMWPE) films[21,22] with higher crystallinities and less entangled amorphous regions compared to prior samples has led to reports of high thermal conductivities exceeding $60\,W m^{-1}\,K^{-1}$ (refs. [23,24]). Several recent studies have also reported high thermal conductivity up to $20-30\,W m^{-1}\,K^{-1}$ in a diverse set of polymers besides PE, including polybenzobisoxazole[18], polyethylene oxide[25], and amorphous polythiophene[26].

Knowledge of the structural changes that occur upon drawing from nascent PE aid in identifying the origin of the high thermal conductivity values, and extensive studies have characterized the atomic and nanoscale structure of PE films at different DR. The nascent structure consists of spherulites[27] which are in turn composed of unoriented stacked lamella in which folded chains are bridged by intra- and inter-lamella tie molecules[28]. The initial crystalline fraction is on the order of $\sim 60-70\%$, as measured using nuclear magnetic resonance (NMR)[29] or heat capacity measurements[30], and from SAXS the crystalline domains have long periods $\sim 10-30\,nm$[31–33] with the corresponding size of the crystalline domain inside the unit being around 90% of the long period[34]. On drawing, Peterlin proposed a sequence of processes occur in which stacked lamella transition to micro-fibrils and eventually to chain-extended crystals[35]. More precisely, on initial drawing, the lamellae begin to align and a crystalline micro-fibril structure bridged by amorphous domains or tie molecules emerges as the lamellae are fragmented. Subsequently, for $10 \lesssim DR \lesssim 50$ the micro-fibrils aggregate with concurrent tautening of the tie-molecules and marginal changes in the crystallinity. Finally, for DR > 50, chain extension leads to an extended crystal phase formed from the aggregated micro-fibrils and tie-molecules. The density of states and dispersion of atomic vibrations in the crystalline phase have been characterized by various inelastic scattering techniques[36–42].

Experimental evidence in support of the above picture has been obtained using various techniques such as transmission electron microscopy (TEM)[43,44], SAXS[45], wide angle x-ray scattering (WAXS)[45], and NMR[46,47], among others[41,48]. For instance, the formation of the micro-fibrils via lamellae fragmentation is consistent with a lack of a clear trend of crystallite size with DR for $DR \lesssim 10$ (refs. [45,49]). The subsequent unfolding and tautening of tie-molecules along with the aggregation of microfibrils is consistent with an initial rapid increase in crystallinity, elastic modulus and orientation factor with DR below DR 20 (ref. [50]) followed by a marginal increase of only a few percent up to DR as large as 200 (refs. [45,50,51]). Evidence for the existence of extended crystal was obtained using various complementary methods such as SAXS and WAXS[47,49,52], TEM[43,44], and NMR[46]. These various techniques indicated that the diameter of the extended crystal is $\sim 10-20\,nm$ and of longitudinal dimension around 100–250 nm or greater[47].

Morphological changes under drawing identified from the above structural studies have provided insight into origin of increase in thermal conductivity. Below DR $\sim 50$, the uniaxial thermal conductivity is observed to monotonically increase with DR[11,53]. Various effective-medium type models have been proposed to interpret this increase in terms of changes in the crystallinity and crystalline orientation[54–57]. Although these models are generally successful in explaining measured thermal conductivity data, the actual transport processes may differ from those assumed by effective medium theory because of the presence of phonons that are ballistic over multiple crystallites. Evidence of such processes has been reported even in partially oriented PE samples with low DR $\sim 7.5$ using transient grating spectroscopy (TG)[58].

For disentangled UHMWPE of DR $\gtrsim 150$, the increase in thermal conductivity is difficult to interpret using the above models because the thermal conductivity is observed to increase on average by factor of $\sim 20\%$ despite a lack of detectable change in crystallinity or chain orientation[11–13,23,24]. Conflicting explanations have been proposed to account for these observations. For instance, Xu et al. used the isotropic helix-coil model to conclude that the thermal conductivity of the amorphous phase ($\kappa_a$) must be as high as $16\,W m^{-1}\,K^{-1}$ to explain the high thermal conductivity of $\sim 60\,W m^{-1}\,K^{-1}$ for samples with DR $\sim 100$ (ref. [24]). On the other hand, Ronca et al. used the same model to conclude that the high thermal conductivity for DR $\gtrsim 150$ originates from the enlargement of the extended crystal dimensions[23]. The discrepancy is difficult to resolve by bulk thermal conductivity measurements because the properties of the crystalline and amorphous phases cannot be independently measured. As a result, the physical origin of the high thermal conductivity of disentangled UHMWPE remains unclear.

Here, we report measurements of the thermal conductivity and mean free path accumulation function of disentangled UHMWPE films (DR $\sim 200$) using cryogenic thermal conductivity measurements and transient grating spectroscopy. The thermal conductivity exhibits a marked grating dependence, indicating the presence of ballistic heat-carrying atomic vibrations over the length scale of a grating period. We interpret the TG and cryogenic thermal conductivity measurements using an anisotropic Debye model that describes heat transport by longitudinal acoustic atomic vibrations. The analysis indicates that the heat is nearly entirely carried by this branch, with values of the temperature-independent mean free paths being around 400 nm up to several THz. Comparing these results to those of our prior study of disentangled UHMWPE films of lower DR[58], we find that the high thermal conductivity for DR $\gtrsim 150$ can be attributed to the presence of enlarged extended crystals. As the phonon MFPs appear to be limited by the dimensions of the extended crystals, our study suggests that disentangled UHMWPE films with higher thermal conductivity may be realized in samples with larger extended crystals.

## Results

**Transient grating spectroscopy on UHMWPE films**. We measured the in-plane thermal conductivity of disentangled UHMWPE films using TG, as schematically illustrated in Fig. 1A. The samples are disentangled UHMWPE films with draw ratio (DR) of 196 (see "Methods"). Figure 1B shows an optical image of the film that is of centimeter scale dimension laterally and of thickness around 30 µm, as measured using calipers. A scanning electron microscope (SEM) image is given in Fig. 1C. In both images, highly oriented fibers extending over tens of microns are visible. Since neat PE is transparent to visible light, Au nanoparticles (diameter: $\sim 2-12\,nm$[59]; concentration: 1 wt%) were

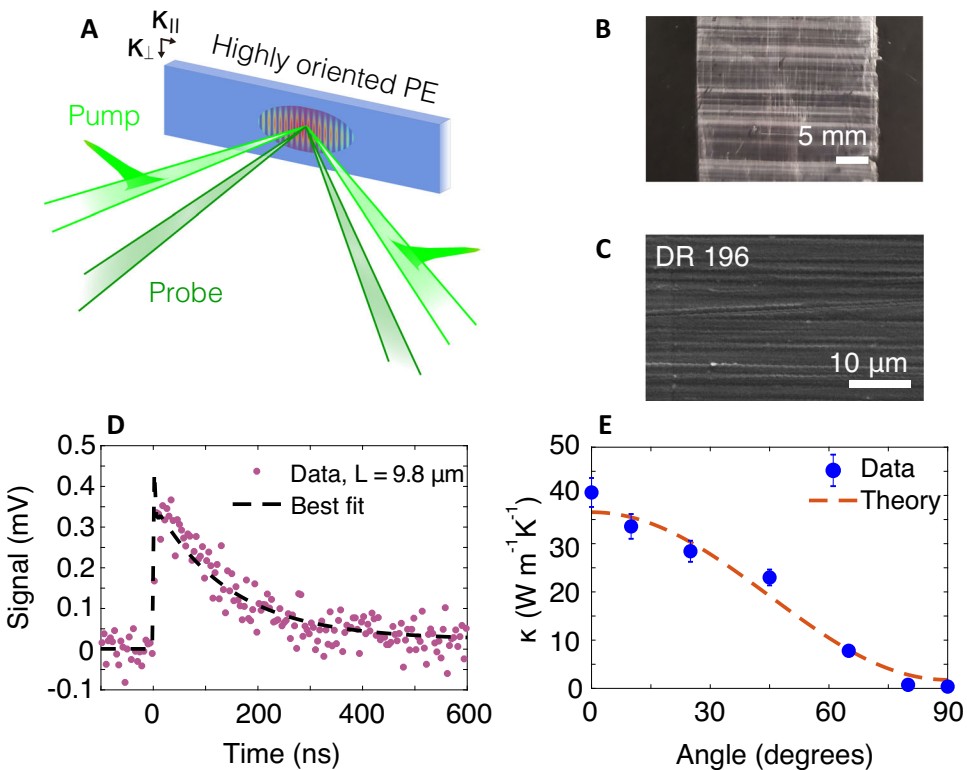

**Fig. 1 In-plane thermal characterization of disentangled UHMWPE films using TG. A** Schematic illustration of transient grating formation and temperature profile. Pump laser pulses impulsively generate a spatial grating on the sample from which probe beams diffract. **B** Optical image of disentangled UHMWPE film (DR = 196). **C** SEM image of the disentangled UHMWPE film. Extended fibers over tens of μm are visible. **D** Representative TG signal versus time for grating period $L = 9.8\,\mu m$ at 300 K. The signal is an average of $3 \times 10^4$ repetitions at a single location on the sample. The measurement was conducted at multiple locations (see Supplementary Material Section 1 for additional data). The thermal diffusivity is obtained as the time constant of the exponential decay. **E** Thermal conductivity versus angle between draw direction and thermal gradient defined by the grating $L = 9.8\,\mu m$. The 0° (90°) indicates heat flow direction parallel (perpendicular) to the draw direction. The maximum thermal conductivity is around 40 W m$^{-1}$ K$^{-1}$ along the chain, while the perpendicular value is comparable to that of unoriented PE.

added as an optical absorber. The concentration of the Au was selected to minimize the effect of the filler on the thermal conductivity while enabling the formation of a thermal grating on the sample[58]. Experimental characterization of similar samples using polarized light microscopy, among other methods, indicates that the nanoparticles are oriented in linear chains in the amorphous regions[59]. A rough estimate of the absorption depth from UV–VIS spectra yields a value of 30–40 μm[59], indicating that the optical absorption depth is on the order of the sample thickness. Considering that the grating periods used in the present work are around an order of magnitude smaller than the cross-plane length scales, one-dimensional heat transfer was assumed in the interpretation of the TG signal.

A representative TG signal measured at grating period $L = 9.8\,\mu m$ is shown in Fig. 1D. As described in the Supporting Information of ref. [58], the signal consists of an initially fast decay (time constant $\lesssim 1\,ns$) followed by a slower decay (time constant $\gtrsim 10\,ns$). The initial fast decay is attributed to the thermal relaxation of the Au nanoparticles, while the subsequent slower decay corresponds to thermal conduction in the film. Following the procedure in ref. [58], we fit the signal with a multi-exponential function; the time constant of the slower decay yields the thermal diffusivity of the sample. Because the initial signal from the nanoparticles exhibits a short time constant compared to their thermal signal, the influence of the nanoparticle signal on the fitted thermal diffusivity is negligible. The signal-to-noise ratio (SNR) of the present measurements is generally <20, which is about 30% of that reported in ref. [58] for DR = 7.5 samples. This

decrease is because the highly oriented samples scatter visible light intensely owing to the increased inhomogeneity over length scales comparable to the optical wavelength, as evidenced by AFM images perpendicular to the fiber alignment direction (see Supplementary Material Section 6 and Supplementary Fig. S5). Nevertheless, TG is able to measure the thermal signal with adequate SNR because only the diffracted signal due to the spatial refractive index profile at the grating wave vector is measured, and the scattered light intercepted by the detector that does not arise from diffraction can be subtracted from the final signal using a heterodyning procedure[60]. The thermal conductivity was calculated from the measured thermal diffusivity using the heat capacities of linear PE in ref. [61].

**Thermal conductivity–temperature dependence.** The in-plane thermal conductivity versus angle between the fiber alignment direction and the thermal gradient is shown in Fig. 1E. The thermal conductivity is ~40 W m$^{-1}$ K$^{-1}$ at 0° ($\kappa_\parallel$, parallel to grating), and decreases to 0.4 W m$^{-1}$ K$^{-1}$ at 90° ($\kappa_\perp$, perpendicular to grating). The value along the draw direction is in reasonable agreement with that obtained on a sample without Au nanoparticles using the laser flash method in ref. [23], indicating that the 1 wt% of AuNPs does not measurably affect the thermal transport properties of the present sample. The value of $\kappa_\perp \sim 0.4\,W\,m^{-1}\,K^{-1}$ is close to that of unoriented PE, which is attributed to the heat conduction by interchain van der Waals interactions[62]. The angle-dependent thermal conductivity was fitted by a geometric model[63] with the thermal conductivity along

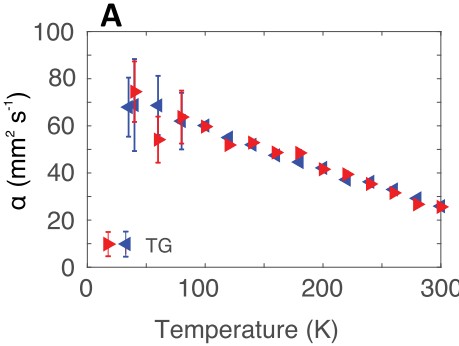
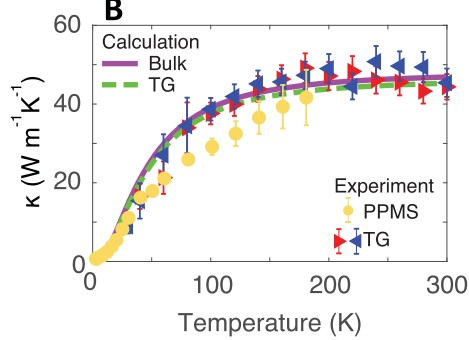

**Fig. 2 Temperature dependence of thermal diffusivity and thermal conductivity. A** Thermal diffusivity along the chain axis versus temperature for grating period $L = 9.8\,\mu m$ (heating: red right-point triangles; cooling: blue left-point triangles). An increase in the thermal diffusivity is observed as temperature decreases. **B** Thermal conductivity along the chain axis versus temperature measured by TG for $L = 9.8\,\mu m$ (heating: red right-point triangles; cooling: blue left-point triangles) and PPMS (yellow circles). The thermal conductivity is approximately constant from ~300−220 K, below which the thermal conductivity decreases. The trend and corresponding values from PPMS and TG with $L = 9.8\,\mu m$ are in reasonable agreement, suggesting that phonon mean free paths are less than ~$L/2\pi$~1.5 μm. Calculated thermal conductivity versus temperature obtained using Eq. (1) (solid purple line: bulk; dashed green line: $L = 9.8\,\mu m$).

the two principal directions as input. Good qualitative agreement between the model and the data is observed.

The thermal transport properties can be further examined by measuring the temperature dependence of the thermal diffusivity and conductivity. The bulk thermal diffusivity along the chain direction versus temperature obtained from TG with $L = 9.8\,\mu m$ between 30 and 300 K is shown in Fig. 2A. The diffusivity exhibits a linearly increasing trend with decreasing temperature, a qualitatively similar trend as that reported in microscale crystalline fibers[14]. The corresponding $\kappa_\parallel$ versus temperature is shown in Fig. 2B. Within the uncertainty of the measurement, the thermal conductivity remains constant from room temperature to ~220 K, below which the thermal conductivity decreases.

The macroscopic dimensions of the present samples permits additional characterization of the thermal conductivity down to ~3 K using a physical property measurement system (PPMS) (see the "Methods" section and Supplementary Material Section 4 for further details). The results are shown in Fig. 2B. The measured values and the trend of the bulk thermal conductivity are consistent with that obtained from TG. We note that the PPMS thermal conductivity represents an average value over a larger spatial dimension than that in TG (~5 mm sample length in PPMS versus 500 μm beam diameter in TG), which may account for the slightly lower values obtained from PPMS at around 100 K. The cryogenic thermal conductivity values on a logarithmic scale are given in Fig. 3. The values exhibit two distinct temperature dependencies with a transition at around 10 K.

The temperature dependence of the thermal diffusivity and conductivity provides insight into the origin of the high thermal conductivity in the present samples. First, the thermal diffusivity is observed to depend on temperature, ruling out a constant relaxation time for all phonon polarizations as suggested in ref. [20]. Second, within the uncertainty of the measurements, the measured bulk thermal conductivity is in reasonable agreement with the TG data for $L = 9.8\,\mu m$, indicating that the phonon mean free paths are shorter than ~$9.8/2\pi$~1.5 μm (ref. [64]). Third, the measured trend of thermal conductivity versus temperature is consistent with structural scattering being the dominant scattering mechanism. Above 10 K, the trend is qualitatively similar to those reported previously for PE films of various DR as shown in Fig. 3, although the thermal conductivity of the present sample is consistently higher. Below 10 K, a weaker trend with temperature is observed compared to those exhibited by other samples. A possible origin of this change in trend is the increased importance

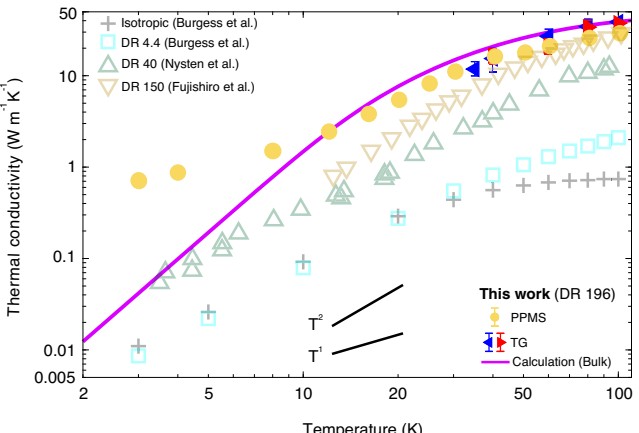

**Fig. 3 Cryogenic bulk thermal conductivity versus temperature below 100 K.** Measurements from this work (TG: filled colored triangles; PPMS: yellow circles) along with the predictions of an anisotropic Debye model (purple solid line) are shown. Representative literature data for semi-crystalline PE with various DRs are also plotted as open symbols (extruded thin film, DR 4.4, ref. [8]; solution-cast thin film, DR 40, ref. [73]; solution-cast macroscopic fiber, DR 150, in ref. [80]). As temperature decreases, the trend of measured thermal conductivity exhibits a transition from ~$T^2$ to ~$T$ near 10 K.

of other phonon branches to heat conduction at cryogenic temperatures. As the heat capacity of the LA branch decreases with decreasing temperature, other branches with smaller group velocities may begin to contribute to thermal conductivity owing to their larger heat capacity, even if their mean free paths are so short as to provide negligible contribution at higher temperatures. At the same time, the scattering mechanisms of these branches may differ from those we have inferred for the L branch, thereby giving a qualitatively different temperature dependence. This hypothesis will be the topic of future study.

**Thermal conductivity—grating period dependence.** Despite these constraints from the temperature-dependent bulk thermal transport properties, the microscopic properties of the heat-carrying atomic vibrations remain underdetermined. To gain further insight, we exploited the ability of TG to systematically vary the induced thermal gradient over micrometer length scales by tuning the grating

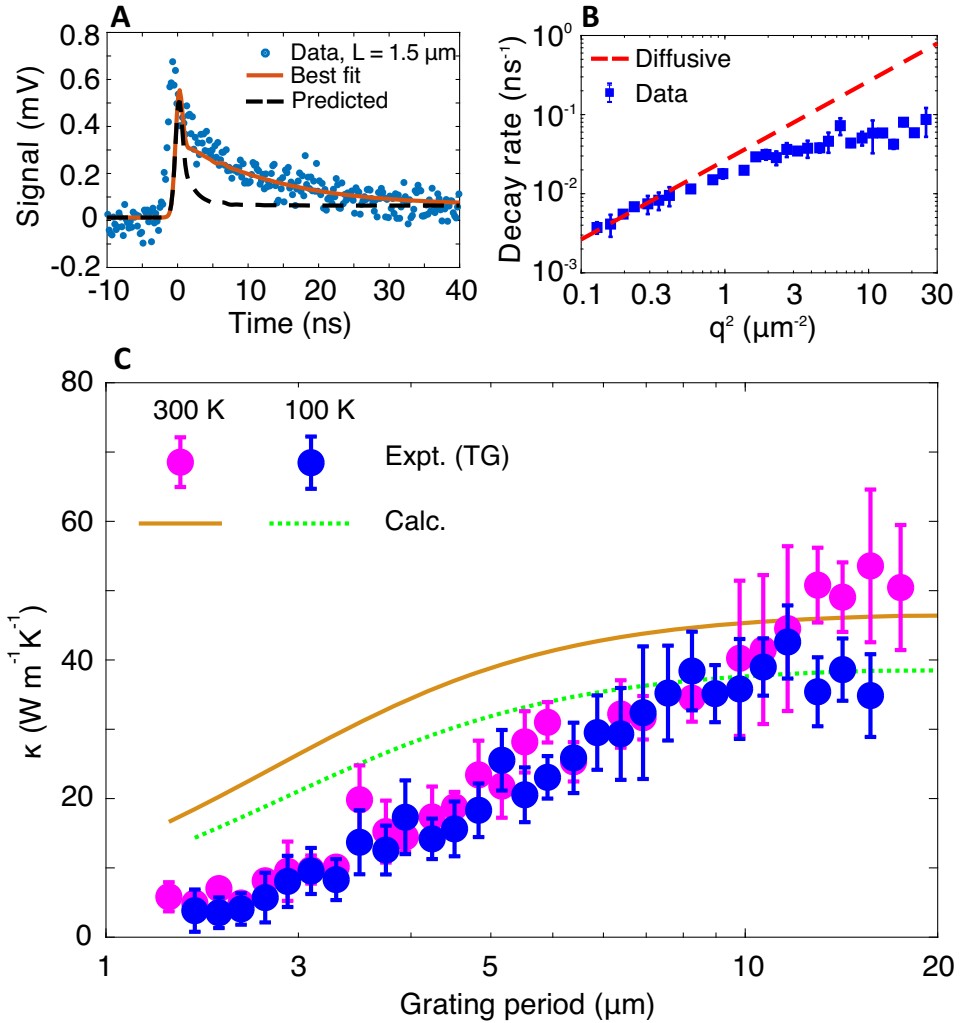

**Fig. 4 Quasiballistic thermal transport in disentangled UHMWPE films. A** Measured TG signal versus time (symbols) for grating period $L = 1.5\,\mu m$, corresponding to $q^2 \sim 17.5\,\mu m^{-2}$, along with the best fit (solid red line) and predicted decay estimated using the thermal conductivity obtained for $L = 9.8\,\mu m$ (dashed black line). The signal is an average of $5 \times 10^4$ shots measured at a single location. The signal clearly decays slower than predicted based on the thermal conductivity measured at larger grating period, indicating a departure from diffusive thermal transport. **B** TG signal decay rate versus $q^2$. The measured decay rates for $q^2 \gtrsim 0.3\,\mu m^{-2}$ deviate from that predicted from the thermal conductivity measured at $L = 9.8\,\mu m$. **C** Experimental thermal conductivity versus grating period at selected temperatures from experiments (magenta symbols: 300 K; blue symbols: 100 K) along with the calculation (orange solid line: 300 K; green dotted line: 100 K). The thermal conductivity exhibits a marked dependence on grating period and is nearly independent of temperature. Error bars indicate 95% confidence intervals obtained using the procedure given in ref. [58].

period. If heat-carrying phonons propagate ballistically over the grating period, the thermal decay is slower than that predicted from the bulk thermal conductivity[60,64]. We have previously used this approach to identify ballistic phonons over nanocrystalline domains in disentangled UHMWPE samples of lower DR[58].

We apply this approach to the present samples, measuring the thermal conductivity along the chain axis versus grating period at temperatures of 300, 220, 100, and 35 K. The measured TG signal for a grating period of $L = 1.5\,\mu m$ is shown in the inset of Fig. 4A. The decay is clearly slower than expected based on the bulk thermal conductivity value, indicating the presence of ballistic phonons on the length scale of the grating period. Measurements of the decay rate versus $q^2$ for all the grating periods at 300 K are given in Fig. 4B. The measured decay rate is close to that predicted by the bulk thermal conductivity at $q^2 \leq 0.3\,\mu m^{-2}$, above which the decay rate is slower by up to a factor of 5 at the smallest grating period.

The corresponding thermal conductivity versus grating period at 300 and 100 K is shown in Fig. 4C. The thermal conductivity

exhibits a marked dependence on grating period up to $\sim$10 μm, a value comparable to that observed in other covalent crystals with higher thermal conductivity such as silicon[60,65]. Compared to PE samples of lower DR $\sim$ 7.5 (ref. [58]), the observed trend is significantly more pronounced, indicating the presence of heat-carrying phonons with longer MFPs in the present samples. The observed grating dependence lacks a clear temperature dependence for the temperatures considered here (see Supplementary Material Section 2 for additional data). This finding indicates the dominance of the structural scattering, consistent with the trend of the bulk thermal conductivity versus temperature in Fig. 2B.

**Low energy anisotropic Debye model**. We now construct a model to interpret the measurements in Figs. 2B, 3, and 4. First, extensive structural characterization of highly drawn samples indicates that the crystallinity of samples with draw ratio around 200 is nearly 90%[47,59], and that the enlarged crystals with a dimension of $\sim$400−500 nm are separated by fragmented

inter-crystalline bridges[44,46,47]. Therefore, we treat the heat-carrying atomic vibrations as the usual phonons in a crystal, with the disordered regions serving as pertubations that induce scattering. We note that considering the complex structure of the polymers, non-propagating atomic vibrations, which are not included in the model, may make a contribution to thermal transport, particularly at cryogenic temperatures[66]. Next, Fig. 4 indicates that phonons with MFPs on the order of hundreds of nanometers carry the majority of the heat. Phonons with such long MFPs are likely from the LA branch owing to its high group velocity $v_c \sim 16 - 17 \text{ km s}^{-1}$ (refs. [38–40,42,67]). For other branches to contribute, their lifetimes would need to be orders of magnitude larger than those of the LA branch to compensate their low group velocity; such long lifetimes are inconsistent with the findings of ab-initio studies[68]. Therefore, the marked dependence of grating period in Fig. 4 implies that nearly all heat is carried by the LA branch. Finally, the grating-period dependent thermal conductivity in Fig. 4C exhibits only a weak temperature dependence, indicating that any temperature-dependence of the MFPs can be neglected at first order.

We therefore use an anisotropic Debye model[69,70] to calculate the heat conducted by this branch and use the data to constrain the frequency-dependence of the LA branch relaxation time. The marked elastic anisotropy of PE can be accounted for to good approximation by assuming the group velocities all point along the chain axis[15]. Note that although the longitudinal density of states is nearly one-dimensional at a sufficiently high frequency, at low frequencies below a few THz this approximation is poor, necessitating the use of the anisotropic Debye model. Considering this discussion, the thermal conductivity measured in TG can be expressed as

$$\kappa_i = \int_0^{\omega_{ab}} S(x_i) \left[ C_1(\omega) v_c \Lambda(\omega) \right] \mathrm{d}\omega + \int_{\omega_c}^{\omega_{ab}} S(x_i) \left[ C_2(\omega) v_c \Lambda(\omega) \right] \mathrm{d}\omega$$

(1)

where $\Lambda(\omega)$ is frequency-dependent mean free path, $q_i = 2\pi L_i^{-1}$ is grating wave vector used for measurement $i$, $x_i = q_i \Lambda(\omega)$, $S(x_i)$ is the anisotropic phonon suppression function for an arbitrary phonon dispersion[71], and $C_1(\omega)$ and $C_2(\omega)$ refer to the heat capacity terms in Eq. (11b) of ref. [70].

Numerical values for the model are obtained as follows. The c-axis velocity of the longitudinal polarization is $v_c \sim 17 \text{ km s}^{-1}$ (refs. [38–40]). The velocity along a perpendicular crystal axis was reported as $v_{ab} \sim 1.35 \text{ km s}^{-1}$ (ref. [38]) from inelastic neutron scattering and $\sim 4.5 \text{ km s}^{-1}$ from an ab-initio calculation[68] which may be reflective of the monocrystals present in disentangled UHMWPE[21]. We roughly estimate $v_{ab} \sim 3 \text{ km s}^{-1}$. We consider $\omega_c$ to be a characteristic frequency at which the c-axis longitudinal velocity decreases below the Debye velocity and ultimately tends towards zero. Roughly, we estimate $\omega_c \sim 10 \text{ THz}$; the analysis below is not sensitive to this choice. This choice then determines $\omega_{ab} = 1.8 \text{ THz}$. The corresponding maximum wave vector magnitude is $\sim 6 \text{ Å}^{-1}$.

We next seek to identify the function $\Lambda(\omega)$ that best explains the temperature and grating dependence of the thermal conductivity. To constrain the MFP function, we note that at $\sim 1 \text{ THz}$, the magnitude of the MFP can be estimated using the dominant phonon approximation[72] and the cryogenic thermal conductivity measurements in Fig. 3. We use the cryogenic thermal conductivity at around 12 K, the minimum temperature that is still on the $T^2$ trend of the thermal conductivity. We find $\Lambda \sim \kappa / C_l v_l \sim 340 \text{ nm}$ at $\sim 1 \text{ THz}$, where $\kappa \sim 2.45 \text{ W m}^{-1} \text{ K}^{-1}$ and $C_l \sim 4.2 \times 10^{-4} \text{ J cm}^{-3} \text{ K}^{-1}$ is the computed heat capacity of the longitudinal branch at 12 K.

With these values specified, the MFP profiles versus frequency that yield the best agreement with the experiments in Figs. 2B, 3,

and 4 are obtained by adjusting the MFP profile subject to the above constraints. We construct MFP profiles using Matthiessen's rule by combining a low frequency constant mean free path ($\Lambda_0$) with power laws at higher frequency, yielding a mean free path function of the form $\Lambda(\omega)^{-1} \sim \Lambda_0^{-1} + \beta \omega^n$. The low frequency value $\Lambda_0 = 340 \text{ nm}$ was used based on the estimate from the cryogenic thermal characterization. The value of the coefficient $\beta$ was obtained by optimizing the fit of the calculated thermal conductivity with the measured TG and PPMS data. After extensive comparison, we found that the best fit is obtained using a constant $\Lambda_0 = 340 \text{ nm}$ up to $\sim 5.5 \text{ THz}$, beyond which the MFP decreases as $\sim \omega^{-4}$, although similar power laws also yield similar qualitative agreement (see Supplementary Material Section 3 for results using other candidate profiles).

The resulting computed bulk thermal conductivity using this profile is presented in Figs. 2B and 3. In Fig. 2B, the bulk thermal conductivity exhibits qualitative agreement with the measurements, producing the observed magnitude and trend. The calculated thermal conductivity in TG with $L = 9.8 \text{ μm}$ is close to the bulk value, consistent with the good agreement between the TG and PPMS measurements. The grating period dependence of the thermal conductivity is also qualitatively reproduced by the calculation in Fig. 4C, and Fig. S2 in Supplementary Material Section 2, although quantitative disrepancies exist. In particular, the dependence of the thermal conductivity versus grating period on temperature predicted by the model is difficult to discern considering the experimental uncertainty and will be the topic of future study.

The calculated cryogenic thermal conductivity is shown in Fig. 3. Although a similar qualitative trend is observed above $\sim 10 \text{ K}$, the agreement is worse below this temperature. This discrepancy could be attributed to heat conduction by other types of atomic vibrations which may make the primary contribution to thermal conductivity below $\sim 10 \text{ K}$.

## Discussion

We now discuss our findings in context with prior studies of thermal conduction in oriented PE films. We first consider the MFP value of 340 nm for frequencies in the THz range. For frequencies around 1 THz, prior values inferred from literature data using the dominant phonon approximation at cryogenic temperatures are $\sim 80 \text{ nm}$ using $\kappa \sim 0.34 \text{ W m}^{-1} \text{ K}^{-1}$ for DR 40 (ref. [73]), as estimated using LA specific heat of $\sim 2.8 \times 10^{-4} \text{ J cm}^{-3} \text{ K}^{-1}$ at 10 K using the model described above. This value is comparable to an estimated value of $\sim 60 \text{ nm}$ for DR 6 at 10 K (ref. [16]). At higher frequencies $\sim 6 \text{ THz}$, IXS has been used to obtain a MFP of $\sim 50 \text{ nm}$ for DR 5.5 (ref. [42]), which is in reasonable agreement with the values inferred from transport studies. The MFPs obtained from TG measurements on a DR 7.5 sample in our previous study ranged from 10 to 200 nm (ref. [58]). All of these values are comparable to but distinctly smaller than the present value, expected as the present samples exhibit higher thermal conductivity. The decrease in MFP at frequencies exceeding $\sim 5.5 \text{ THz}$ is also consistent with the increase in broadening reported in IXS[42], although this increase occurred at higher frequencies in their data.

Our findings help to explain the origin of high thermal conductivity in disentangled UHMWPE films. Recent studies have provided conflicting explanations for the high thermal conductivity, with Xu et al. [24] attributing it to the high thermal conductivity of the amorphous phase[58] but Ronca et al. [23] attributing it to increased extended crystal dimensions. Our data and analysis are consistent with the latter explanation. Compared to the MFPs of a DR 7.5 sample in our prior study[58], the present MFPs are clearly larger, as would be expected if the extended

crystal dimensions have increased. The crystal dimensions may increase without increasing the crystallinity by merging of smaller crystals. The MFPs in both samples exhibit a clear maximum in the low THz frequencies and are independent of temperature, suggesting that phonon scattering is predominantly due to reflections at crystalline domain boundaries. Further, evidence exists for the presence of extended crystals with length on the order of hundreds of nanometers from NMR and TEM[44,46,47]. These values are compatible with the inferred value of $\Lambda_0$. We note that $\Lambda_0$ should be interpreted as an average dimension of the crystallites for phonon scattering, which may not coincide precisely with their actual physical dimension depending on the phonon transmission coefficient across the boundary.

We infer that heat conduction in disentangled UHMWPE is due to longitudinal atomic vibrations that are ballistic within the extended crystal phase, scattered primarily by reflections at the boundaries between the crystals. Our results therefore support the hypothesis of ref. [23] in which the high thermal conductivity compared to the prediction including only crystallinity and anisotropy factors for DR $\gtrsim$ 180 (Fig. 7 in ref. [23]) was attributed to the enlargement of extended crystal dimensions. We note that while other mechanisms such as molecular conformation disorder have been proposed to affect the thermal conductivity of crystalline polymers[74], these mechanisms do not appear to exhibit the dependence on draw ratio required to explain the results of this study and ref. [58]. We conclude that the increase of crystalline dimensions is the explanation that best accounts for the reported experimental data.

Finally, we discuss the implications of our findings for realizing PE films of higher thermal conductivity. Because the MFPs appear to be limited by the size of the extended crystals, our study indicates that the thermal conductivity of PE films has not yet reached its upper limit. The practical challenge is synthesizing disentangled UHMWPE films with larger extended crystal dimensions. Such films would be expected to have higher thermal conductivity of an amount proportional to the increase in crystalline dimension.

## Methods

**Sample preparation**. The disentangled UHMWPE thin films were synthesized using the same procedure given in ref. [21,59], but with higher draw ratio (DR = 196) achieved by rolling (×7) and stretching (×28). The average molecular weight ($M_w$) was characterized to be $M_w = 5.6 \times 10^6$ gmol$^{-1}$ (ref. [59]), as measured using rheological measurements.

The synthesis procedure was as follows. First, dodecanethiol-functionalized Au nanoparticles (quoted diameter distribution: 2–5 nm) were purchased from Sigma-Aldrich dissolved in toluene (2% w/v) and used as received. Then, powdered disentangled UHMWPE was suspended in acetone and magnetically stirred for 1 h. Subsequently, the solution consisting of dodecanthiol-Au nanoparticles dissolved in toluene was added into the dis-UHMWPE/acetone suspension under magnetic stirring overnight and then further dried at 50 °C for 1 h to completely evaporate both solvents, while not compromising the disentangled nature of the polymer[75]. The Au NPs concentration was constant at 1.0 wt%. The dried nanocomposite powder was compression molded at 125 °C (solid-state) to produce slabs and then drawn 7 times from their initial length via twin-roll calendering (speed: 0.1 rpm; temperature: 125 °C)[59,76]. Finally, the specimens were additionally drawn by ×28 to achieve DR = 196 under tensile stretching (speed: 50 mm per minute in Hounsfield tensometer; temperature: 125 °C)[59].

**Transient grating spectroscopy**. The TG setup employed in this work is identical to that described in ref. [58]. Briefly, a pair of pump pulses (wavelength 515 nm, beam diameter 530 μm, pulse duration ~1 ns, pulse energy 13 μJ, repetition rate 200 Hz) is focused onto the sample to impulsively create a spatially sinusoidal temperature rise of period $L$ and wave vector $q = 2\pi L^{-1}$. The grating relaxes by heat conduction, and its decay is monitored by the heterodyne measurement of a diffracted CW signal beam and reference probe beam (wavelength 532 nm, beam diameter 470 μm, CW power 900 μW, chopped at 3.2% duty cycle to reduce steady heating on the sample).

**Physical property measurement system**. We performed bulk thermal conductivity measurements at cryogenic temperatures using a commercial 7 T Dynacool Physical Property Measurement System (PPMS, Quantum Design). Samples of

196 DR (thickness: 30 μm, as measured using the calipers and cross-sectional view from scanning electron microscopy; lateral width: 1.62 mm; heat conduction length excluding electrical contact: 5.6 mm) were mounted in a four-point electrical contact geometry. Silver conducting epoxy was applied between the sample and the four copper wires and cured for 7 h at ~400 K on a hot plate. Following refs. [77,78], we additionally applied a cryogenic varnish on top of the epoxy (GE7031, Lakeshore) to ensure both physical and thermal contact between the wires and the sample; the varnish was cured at room temperature for 24 h and then transferred onto a thermal transport platform. Heat capacity measurements were attempted but were not successful owing to experimental difficulties in mounting the sample. Therefore, heat capacities from ref. [61] were used. Although our samples contained Au nanoparticles, we here note that the heat capacities of gold on a mass basis are smaller by a factor of 4–10 compared to that of the PE (at 300 K, 0.13 in ref. [79] vs. 1.75 J g$^{-1}$ K$^{-1}$ in ref. [61]). The contribution of 1 wt% of the Au to the total heat capacity in our sample will therefore be negligible.

**Reporting summary**. Further information on research design is available in the Nature Research Reporting Summary linked to this article.

## Data availability
The authors declare that the data supporting the findings of this study are available within the paper and its Supplementary Material files.

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

## Acknowledgements

The authors thank Bolin Liao and Wenkai Ouyang for assistance with PPMS measurements. This work was supported by the Office of Naval Research under Grant Number N00014-18-1-2101.

## Author contributions

T.K. and A.J.M. conceived the project. S.X.D. and S.R. fabricated the samples. T.K. performed the TG and PPMS measurements and analyzed the thermal measurements data. All authors discussed the results. T.K. and A.J.M. wrote the manuscript with contributions from all authors.

## Competing interests

The authors declare no competing interests.
