## [Peer review file · Nature Communications]

REVIEWER COMMENTS

Reviewer #1 (Remarks to the Author):

In this submission, the authors applied a transient grating (TG) technique to elucidate the microscopic origin of high thermal conductivity of the disentangled ultrahigh molecular weight polyethylene (UHMWPE) films. The topic of this work should be of interest to those researchers who work on the area of thermally conductive polymers. Nevertheless, I feel there are several points to be clarified before this manuscript can be considered further for Nat. Commun.

1) In this manuscript, the in-plane thermal conductivity was measured on the ~30 micrometer-thick sample, on Page 4, II EXPERIMENT. I am not quite sure whether the sample thickness is too large. For the in-plane thermal conductivity measurement based on the TG technology, the light adsorption depth at the pumping wavelength should be far larger than the film thickness to ensure one-dimensional in-plane heat transfer. One can refer to Johnson et al, PRL 110, 025901 (2013), Page 2. Is there any means to estimate the light adsorption depth in the sample?

2) I noticed that the thermal conductivity was calculated from measured TG thermal diffusivity using the heat capacity data of linear PE in Ref. 61, Page 6. The sample used in this study is the UHMWPE films incorporated with 1.0 wt% Au NPs. Why the heat capacities of the sample were not specifically measured by the authors to rule out the possible inconsistencies between the linear PE sample and the Au NPs- incorporated UHMWPE films?

3) The thermal conductivities measured by the PPMS were found to be in line with those obtained from TG. Did the sample used for the PPMS measurement also contain the Au NPs? If this is true, it may be a good idea to use the PPMS to characterize a UHMWPE film without adding Au NPs. If the obtained thermal conductivity values agree with those of the Au NPs incorporated sample, it may suggest that the Au-NPs do not significantly affect the heat transfer in the PPMS and TG measurements.

4) Can the authors give an explicit explanation for the transition of the temperature dependent thermal conductivity from $\sim T^2$ dependency to $\sim T$ dependency near 10 K? Should this be attributed totally to the heat capacity change?

5) In Eq.(1), the term $\Lambda(\omega)$ (mean free path function?) was not defined. For the term $\chi_{i,s}$, the letter s was not defined either.

6) Is there a mathematical expression for the MFP profile shown in Figure SIV(A)? How to construct the non-linear part of the MFP profile below the transition frequency, for instance, ~ 7 THz in Fig. SIV(A)?

Reviewer #2 (Remarks to the Author):

In this work, the authors studied a highly oriented polyethylene sample to examine the microscopic mechanism of the high thermal conductivity. An intriguing finding is the large grating period dependence of thermal conductivity observed in transient grating experiment. Heat transport mechanism in polymers is an important topic for both fundamental understanding of heat conduction in disordered materials and practical applications. This work provides evidence about the existence of propagating acoustic modes with propagating lengths up to 10 μ m, and can help others to further discover heat-conducting polymers. However, I am not fully convinced a phonon model would be appropriate in such case. I would like to recommend the publication if the authors can consider the following points.

1. My major question is that while I think it is helpful to think about heat transport using phonon model, I am not convinced it can quantitatively model heat transport in polymers. This

is because while crystalline domains should possess phonon modes, these domains are separated by intermediate regions that are somewhat disordered and do not have rigorous periodicity. If one would like to use phonon model, it will be more appropriate to also incorporate phonon transmission through the intermediate region.

The authors assumed a temperature-independent phonon mean free path and explained this can be rationalized by scattering at the crystalline boundaries. However, in polycrystalline materials the crystalline boundary is nearly 2D compared to crystalline domains. In contrast, in semicrystalline polymers, the domains are separated by disordered regions that have finite width. In such case, the forward transmission might be much smaller than the backward reflection. This makes the usual v/L argument for boundary scattering not quite applicable, and the model presented in Section III less convincing.

2. In Fig. 4c, 300K data is almost identical to 100K data for grating period smaller than 10 μ m. If one compares 300K data with 100K data, it seems to tell that a major portion of vibrational modes have the same propagating lengths and contribute nearly the same to heat transport, but at 300K a new batch of modes (with propagating lengths longer than about 10 μ m) is unlocked that further increases the thermal conductivity. This trend is in fact quite unusual. The model instead predicts the grating period

dependence would shift upwards, which seems not in good agreement with the experimental data. One possibility is that at higher temperatures other modes (diffuson, or locon) are excited that start to bridge the acoustic modes in the crystalline domains. This interpretation however is not accurate either. The authors may want to provide more discussions on the possible explanations.

3. There are wiggles in the fitting curve at short delay times (Fig. S1). Why do the wiggles appear?

Minor comments:

4. On page 6 (first paragraph), the sentences "Figure 1(B) shows..are visible" are repeated.

Reviewer #3 (Remarks to the Author):

This paper studies the thermal conductivity in drawn PE films using a transient thermal grating (TTG) method. But changing the grating period length and temperature, it was concluded that phonons in ultra-drawn, highly crystalline PE films are limited by crystal domain sizes instead of anharmonic scattering seen in conventional 3D crystals. This conclusion is based on the observation that the thermal conductivity variations as a function of the grating period length do not show a significant difference for 300 K vs. 100 K. This is also supported by the temperature trend of the thermal conductivity from 0 to 300 K. Using a model to fit the temperature-dependant thermal conductivity, the authors suggest that longitudinal modes dominate thermal transport in such films and the average phonon mean free path (MFP) is ~ 400 nm. It was concluded that there is still room for further improving thermal conductivity PE films if the crystal domain size can be further increased when more advanced fabrication techniques become available. I believe this paper will be of great interest to the thermal transport community and could potentially stimulate the polymer engineering community to further optimize manufacturing techniques. I recommend the acceptance of this manuscript if the following comments can be addressed:

1- How do the authors know for sure it is the crystal domain boundary scattering that limits the phonon MFP? Can this be verified by some structural characterization, maybe from previous works? Is there any evidence that the crystal domain size is ~ 400 nm or in this order of magnitude? Previous studies have also showed that the cis-trans conformation disorder can greatly impact phonon MFP. Could such conformation disorder be a source of MFP limitation?

2- How to explain the divergence between the 300 K and 100 K data in Fig. 4c? I was thinking the main difference between would be from higher frequency modes excited at higher temperature, but it is hard to explain why the divergence happen at long grating periods.

3- The low energy Debye model analysis would benefit from the scenario if TA or torsional modes are used to do the same calculation, and compared to the one based on LA modes. Given the smaller frequency of the TA and torsional modes, the extracted MFPs will be longer. How would these MFPs compare to crystal domain sizes if such characterization is available.

Revision Report for MS#NCOMMS-22-52086

entitled:

Origin of high thermal conductivity in disentangled
ultra-high molecular weight polyethylene films:
ballistic phonons within enlarged crystals

Taeyong Kim, Stavros X. Drakopoulos, Sara Ronca, and Austin Minnich

February 17, 2022

We thank the reviewers for their time to carefully read and provide thoughtful feedback on the manuscript. We have addressed the reviewers' comments point by point and made substantial changes to the manuscript. We believe that it is markedly improved and is now suitable for publication in Nature Communications.

Note that all citation numbers given below are those of the revised manuscript.

Referee # 1

Comment 1.1: In this manuscript, the in-plane thermal conductivity was measured on the ~ 30 micrometer-thick sample, on Page 4, II EXPERIMENT. I am not quite sure whether the sample thickness is too large. For the in-plane thermal conductivity measurement based on the TG technology, the light adsorption depth at the pumping wavelength should be far larger than the film thickness to ensure one-dimensional in-plane heat transfer. One can refer to Johnson et al, PRL 110, 025901 (2013), Page 2. Is there any means to estimate the light adsorption depth in the sample?

Response 1.1:

We thank the reviewer for identifying this important point regarding the required sample properties for TG. Estimating the optical absorption depth is very challenging with our

samples because they intensely scatter light; therefore, separating the scattered and absorbed light is difficult. In fact, the most effective means to determine if the samples absorb at all is by searching for the TG signal. A rough estimate of the absorption depth from UV-VIS spectra yields an absorption depth of 30 – 40 μm (Fig. 4 in Ref. [60]), indicating that the optical absorption depth is on the order of the sample thickness. Considering that the grating period is around an order of magnitude smaller than the cross-plane length scales, the heat transfer is accurately described as one-dimensional. A discussion of the optical absorption depth regarding its estimated value and the validity of one-dimensional heat transfer have been added near Fig. 1C.

Comment 1.2: I noticed that the thermal conductivity was calculated from measured TG thermal diffusivity using the heat capacity data of linear PE in Ref. 61, Page 6. The sample used in this study is the UHMWPE films incorporated with 1.0 wt% Au NPs. Why the heat capacities of the sample were not specifically measured by the authors to rule out the possible inconsistencies between the linear PE sample and the Au NPs- incorporated UHMWPE films?

Response 1.2:

We thank the reviewer for his/her careful attention to the value of the heat capacity used to compute thermal conductivity. There are two points to consider. First, a key result of our study, which forms the basis of our findings, is the observation of a grating-dependent thermal conductivity, Fig 4c of the manuscript. This trend is unaffected by the choice of heat capacity value. Second, although a measurement of the heat capacity of our samples is desirable, the heat capacities of gold on a mass basis are smaller by a factor of 4 – 10 compared to that of the PE (at 300 K, $0.13 \text{ J g}^{-1}\text{K}^{-1}$ from Tab. 1 in Ref. [80] vs $1.75 \text{ J g}^{-1}\text{K}^{-1}$ from Tab. 1 in Ref. [62]). The contribution of 1 wt% Au to the total heat capacity in our sample will therefore be negligible.

In fact, we did attempt to measure heat capacity using various methods such as differential scanning calorimetry (DSC) and the PPMS, but various experimental complications (e.g. difficulties in mounting the samples) prevented us from doing so. To clarify this point, we have added discussion in the “Physical property measurement system” subsection of the “Methods”.

Comment 1.3: The thermal conductivities measured by the PPMS were found to be in

line with those obtained from TG. Did the sample used for the PPMS measurement also contain the Au NPs? If this is true, it may be a good idea to use the PPMS to characterize a UHMWPE film without adding Au NPs. If the obtained thermal conductivity values agree with those of the Au NPs incorporated sample, it may suggest that the Au-NPs do not significantly affect the heat transfer in the PPMS and TG measurements.

Response 1.3:

We are grateful to the reviewer for this idea. In fact, our measurements did agree with the experimentally measured bulk thermal conductivities reported in a prior work (Ref. [23]) for UHMWPE of similar draw ratio without any nanoparticles or fillers. In the original manuscript, we mentioned: “The value along the draw direction is in reasonable agreement with that obtained on a sample without Au nanoparticles using the laser flash method.” To make the connection to the reviewer’s comment, we have added the phrase: “, indicating that the 1 wt% of AuNPs does not measurably affect the thermal transport properties of the present sample.”

Comment 1.4: Can the authors give an explicit explanation for the transition of the temperature dependent thermal conductivity from T^2 dependency to T dependency near 10 K? Should this be attributed totally to the heat capacity change?

Response 1.4:

We appreciate this question. At this time, we do not have a firm hypothesis as to the origin of this trend, but we can make several observations. We believe that the observed transition in the cryogenic thermal conductivity cannot be fully explained by the heat capacity. From Tab. 1 of Ref. [62], the trend of the specific heat capacity ($C(T)$) versus temperature follows $C(T) \propto T^3$ down to ~ 2 K. A possible alternate explanation is the increased importance of other phonon branches to heat conduction at cryogenic temperatures. As the heat capacity of the LA branch decreases with decreasing temperature, other branches may begin to contribute to thermal conductivity even if their mean free paths are so short as to provide negligible contribution at higher temperatures. The scattering mechanisms of these branches may differ from those we have inferred for the L branch, thereby giving a qualitatively different temperature dependence. We have expanded the explanations in the discussion of

data below 10 K near Fig. 3 to include additional discussion of this point.

Comment 1.5: In Eq.(1), the term $\Lambda(\omega)$ (mean free path function?) was not defined. For the term ξ_s , the letter s was not defined either.

Response 1.5:

We thank the reviewer for pointing out this omission. The mathematical expressions along with the necessary definitions have been carefully checked and corrected.

Comment 1.6: Is there a mathematical expression for the MFP profile shown in Figure SIV(A)? How to construct the non-linear part of the MFP profile below the transition frequency, for instance, ~ 7 THz in Fig. SIV(A)?

Response 1.6:

The MFP profiles were constructed using Matthiessen’s rule by combining a low frequency constant mean free path (Λ_0) with power laws (either $n = 1, 2$, or 4) at higher frequency. The resulting mean free path has an expression of $\Lambda(\omega)^{-1} = \Lambda_0^{-1} + \beta\omega^n$. The value $\Lambda_0 = 340$ nm was obtained by analyzing the cryogenic thermal conductivity measurements using the dominant phonon approximation as described in the text. The value of the coefficient β was obtained by optimizing the fit of the calculated thermal conductivity with the measured TG and PPMS data. We have added the equation for Matthiessen’s rule into the construction of MFP profiles in Sec. “Low energy anisotropic Debye model”, as well as the choice of the frequency power laws into the Supporting Information Sec. “SIII Computed thermal conductivity versus grating period versus temperature using other candidate profiles” for clarification.

Referee # 2

Comment 2.1: My major question is that while I think it is helpful to think about heat transport using phonon model, I am not convinced it can quantitatively model heat transport in polymers. This is because while crystalline domains should possess phonon modes, these domains are separated by intermediate regions that are somewhat disordered and do not have rigorous periodicity. If one would like to use phonon model, it will be more appropriate to also incorporate phonon transmission through the intermediate region.

Response 2.1:

We thank the reviewer for raising this important point. It is certainly true that our samples possess disordered regions separating crystalline domains. At the same time, extensive structural characterization of the samples, fabricated by the same group using the same procedure but with lesser draw ratio relative to present sample, (given in e.g. Fig. 7 in Ref. [60]), indicates that the crystallinity of samples with draw ratio around 200 is nearly 90%, in reasonable agreement with studies by other groups (e.g. Ref. [47]). Additionally, prior works have reported so-called “continuous crystals” that are extended over $\sim 400 - 500$ nm, but connected to each other by fragmented inter-crystalline bridges, as inferred using transmission electron microscopy (TEM in Ref. [44]) and nuclear magnetic resonance (NMR in Refs. [46-47]). Therefore, a reasonable description of heat conduction in the sample would seem to be that of phonons in a crystal with the disordered regions as a perturbation that induces scattering. This indeed is the essence of the anisotropic Debye model we have proposed. The disordered region between crystallites serves to limit the mean free path of phonons within the crystalline phase. Due to the clear grating dependence of the thermal conductivity given in Fig. 4C, we believe that a phonon gas model for the crystallites is necessary to account for the long mean free paths required to produce this trend. We have updated our discussion near the description of our model construction in the Sec. “Low energy anisotropic Debye model”.

Comment 2.2: The authors assumed a temperature-independent phonon mean free path and explained this can be rationalized by scattering at the crystalline boundaries. However, in polycrystalline materials the crystalline boundary is nearly 2D compared to crystalline domains. In contrast, in semicrystalline polymers, the domains are separated by disordered regions that have finite width. In such case, the forward transmission might be much smaller than the backward reflection. This makes the usual v/L argument for boundary scattering not quite applicable, and the model presented in Section III less convincing.

Response 2.2:

We appreciate this observation. The primary effect of a decreased forward transmission in the context of our simple model is to cause the fitted Λ_0 to differ by some amount from the physical crystallite size. Λ_0 thus represents an effective value rather than the precise size of the crystalline domains (which is not known in these samples but expected to be

on a similar order as those reported in Ref. [47]). We expect the qualitative predictions of the model to remain valid despite this adjustment. We have updated the second paragraph of Sec. “Discussion”, to highlight that effective length scale for boundary scattering is the effective length (Λ_0) rather than the precise physical crystalline dimension.

Comment 2.3: In Fig. 4c, 300 K data is almost identical to 100 K data for grating period smaller than 10 μm . If one compares 300 K data with 100 K data, it seems to tell that a major portion of vibrational modes have the same propagating lengths and contribute nearly the same to heat transport, but at 300 K a new batch of modes (with propagating lengths longer than about 10 μm) is unlocked that further increases the thermal conductivity. This trend is in fact quite unusual. The model instead predicts the grating period dependence would shift upwards, which seems not in good agreement with the experimental data. One possibility is that at higher temperatures other modes (diffuson, or locon) are excited that start to bridge the acoustic modes in the crystalline domains. This interpretation however is not accurate either. The authors may want to provide more discussions on the possible explanations.

Response 2.3:

We thank the reviewer for his/her careful inspection of the TG data. We agree that the increase of thermal conductivity versus grating period at around 10 microns at 300 K compared to 100 K does not have an immediately obvious explanation. The model predicts that the 300 K thermal conductivity values should be higher than those of 100 K at all grating periods. We note that on average, the 300 K data are somewhat higher than those of 100 K, but the difference lies within the experimental uncertainty. Given this complication, we hesitate to draw further conclusions regarding the nature of atomic vibrations that are relevant at higher temperatures. We have modified the discussion of the TG data in Fig. 4C to mention this point.

Comment 2.4: There are wiggles in the fitting curve at short delay times (Fig. S1). Why do the wiggles appear?

Response 2.4: We thank the reviewer for his/her careful observation. The reason for the appearance of the wiggles in the fitting curves at shorter time scales are due to the impulse response of the detector. To facilitate comparing the predicted and measured TG signal, we computed the fitting curves using a convolution between the detector’s measured

impulse response and the exponential decay corresponding to the material response. Oscillations due to the impulse response of the detector thus appear at time scales around the inverse bandwidth of the detector. To clarify this point, we have now updated the discussion in the Supporting Information Sec. “SI Additional transient grating data” to include the origin of the wiggles.

Comment 2.5: On page 6 (first paragraph), the sentences “Figure 1(B) shows..are visible” are repeated.

Response 2.5: We appreciate the reviewer for pointing this out. We have carefully checked the paper and removed redundant wordings.

Referee # 3

Comment 3.1: How do the authors know for sure it is the crystal domain boundary scattering that limits the phonon MFP? Can this be verified by some structural characterization, maybe from previous works? Is there any evidence that the crystal domain size is ~ 400 nm or in this order of magnitude? Previous studies have also showed that the cis-trans conformation disorder can greatly impact phonon MFP. Could such conformation disorder be a source of MFP limitation?

Response 3.1: We thank the reviewer for suggesting the alternate explanation of cis-trans conformation disorder for the MFP limitation. We believe that the crystalline domain boundaries limiting the MFP is the explanation that best fits the available evidence. First, a lower bound for the crystalline size can be obtained from our prior study (Ref. [58]), in which SAXS was used to determine a typical crystallite size of ~ 60 nanometers in UHMWPE samples of draw ratio around 10. Second, prior structural studies have indicated that these domains increase on drawing up to a value of around a few hundred nanometers as inferred from e.g. transmission electron microscopy (TEM) and nuclear magnetic resonance (NMR) (c.f. Refs. [44] and [46-47]). Considering these results, it seems reasonable to infer that the domain sizes are on this order in the present samples. This value is also on the order of the inferred mean free paths from the PPMS data as mentioned in the text. As pointed out by the reviewer, we agree that it is certainly possible that molecular conformational disorder along the chain can lead to phonon scattering, resulting in changes in the mean free path along with thermal conductivity (c.f. Zhang et al., *JPCC*, **118**:21148-21159, now

added as Ref. [75]). However, a dependence of the cis-trans disorder on draw ratio, which would be required to explain the present results and those of our prior study Ref. [58], was not reported before to the best of our knowledge. We have added a mention of the cis-trans disorder mechanism to scatter phonons in the third paragraph of Sec. “Discussion” in the main text.

Comment 3.2: How to explain the divergence between the 300 K and 100 K data in Fig. 4c? I was thinking the main difference between would be from higher frequency modes excited at higher temperature, but it is hard to explain why the divergence happen at long grating periods.

Response 3.2:

As mentioned in Response 2.3, this feature lacks an immediately obvious explanation. We refer the reviewer to this response for further discussion. The manuscript has been updated with this discussion.

Comment 3.3: The low energy Debye model analysis would benefit from the scenario if TA or torsional modes are used to do the same calculation, and compared to the one based on LA modes. Given the smaller frequency of the TA and torsional modes, the extracted MFPs will be longer. How would these MFPs compare to crystal domain sizes if such characterization is available.

Response 3.3:

We thank the reviewer for this suggestion. The TG data imply that these other vibrations cannot make a substantial contribution to heat conduction above cryogenic temperatures. Specifically, to explain the marked grating dependence, the dominant mean free paths must be comparable to the grating period. To achieve these long mean free paths, their lifetimes would have to be orders of magnitude larger than those of the LA branch to compensate their low group velocity. There is a lack of evidence for such long lifetimes (c.f. an ab-initio calculation of transverse branch lifetimes in Ref. [70]). Therefore, the LA branch appears to be the dominant contributor to thermal transport above cryogenic temperatures. As mentioned in Response 1.4, it is possible that below 10 K, other branches may contribute. This possibility will be the subject of a future investigation. To clarify this point, we have updated explanation of our hypothesis near the discussion of our TG data in Fig. 4C, along

with the description of the model construction in Sec. “Low energy anisotropic Debye model”.

REVIEWER COMMENTS

Reviewer #1 (Remarks to the Author):

The authors have mostly addressed the reviewers' comments and made an appropriate revision. But by carefully looking into it, it seems to me a few things remain to be clarified. For instance, the differences of PPMS and TG results are notably large in 40-200 K. The temperature dependence of both is largely different. In addition, in Fig. 4c showing grating period dependent thermal conductivities of 100 and 300 K, why the data below 300 K are nearly the same? I do not see any interpretation since the values of 300 K should be greater than those of 100 K.

Reviewer #2 (Remarks to the Author):

The authors have provided additional discussion and the manuscript is much improved. I am in favor of the publication. Nonetheless, I hope the authors can consider adding more discussions about the model to further guide the readers in understanding the heat conduction mechanism.

Specifically, while the revised manuscript now provides the explanation about how phonon gas model approximates the heat conduction in polymer case, given that the polymer structure is much more complex than a polycrystalline material, it would be helpful to mention what aspects are not included by the phonon gas model (e.g. non-propagating modes, and non-conventional transmission at the inter-crystalline regions). This way the readers can appreciate the model better as well as recognize the limitations.

Reviewer #3 (Remarks to the Author):

The revision is satisfactory. I would like to recommend the acceptance of this paper.

Revision Report #2 for MS#NCOMMS-22-52086A

entitled:

Origin of high thermal conductivity in disentangled
ultra-high molecular weight polyethylene films:
ballistic phonons within enlarged crystals

Taeyong Kim, Stavros X. Drakopoulos, Sara Ronca, and Austin Minnich

March 7, 2022

We again thank the reviewers for their time to review the revised paper and our response to their comments. We have addressed the remaining concerns point by point and made changes to the manuscript. We believe that the manuscript is now suitable for publication in Nature Communications.

Referee # 1

Comment 1.1: The authors have mostly addressed the reviewers' comments and made an appropriate revision. But by carefully looking into it, it seems to me a few things remain to be clarified.

Response 1.1: We are grateful to the reviewer for the positive feedback and comments. Below, we have listed our response to the reviewer's comments.

Comment 1.2: For instance, the differences of PPMS and TG results are notably large in 40-200 K. The temperature dependence of both is largely different.

Response 1.2: We thank the reviewer for his/her careful inspection of the data. We note that TG and PPMS characterize thermal transport over significantly different length scales, with the relevant scale being the beam diameter (~ 500 microns) in TG and the

sample length (~ 5 mm) for PPMS. The PPMS data therefore reflect properties that are averaged over a much larger volume and could lead to differences from the values from TG. In addition, we note that some heterogeneity in the samples does exist despite the identical procedures used to fabricate these samples.

Despite this quantitative difference, it does not affect the primary finding regarding the nature of heat-carrying atomic vibrations in these samples. To address the reviewer's concern, we have added a mention of the possible origin of the discrepancy on the description of the TG and PPMS data in Fig. 2B.

Comment 1.3: In addition, in Fig. 4c showing grating period dependent thermal conductivities of 100 and 300 K, why the data below 300 K are nearly the same? I do not see any interpretation since the values of 300 K should be greater than those of 100 K.

Response 1.3: We thank the reviewer for his/her for raising this point. The reviewer is correct in that, according to our model prediction, the 300 K thermal conductivity values should be higher than those of 100 K at all grating periods, but with nearly the same grating period dependence. As we mentioned in our previous response to the reviewer 2 and 3, our careful inspection indicates that, on average, the 300 K data are somewhat higher than those of 100 K, but the difference lies within the experimental uncertainty. Given this complication, we hesitate to draw further conclusions regarding this point. Thus, in, our previous response to the reviewer 2 and 3, we had already added our discussion by stating "The grating period dependence of the thermal conductivity is also qualitatively reproduced ... In particular, the dependence of the thermal conductivity versus grating period on temperature predicted by the model is difficult to discern considering the experimental uncertainty and will be the topic of future study."

Referee # 2

Comment 2.1: The authors have provided additional discussion and the manuscript is much improved. I am in favor of the publication. Nonetheless, I hope the authors can consider adding more discussions about the model to further guide the readers in understanding the heat conduction mechanism. Specifically, while the revised manuscript now provides the explanation about how phonon gas model approximates the heat conduction in polymer case, given that the polymer structure is much more complex than a polycrystalline material, it

would be helpful to mention what aspects are not included by the phonon gas model (e.g. non-propagating modes, and non-conventional transmission at the inter-crystalline regions). This way the readers can appreciate the model better as well as recognize the limitations.

Response 2.1: We thank the reviewer for this suggestion. We now have included a mention that these types of atomic vibrations may make a contribution to heat conduction considering the complex structure of polymers, particularly at cryogenic temperatures, near Sec. “Low energy anisotropic Debye model”. A reference to a review paper on heat conduction in disordered materials (F. DeAngelis et al., *Nanoscale Microscale Thermophys. Eng.*, **23**:81-116, 2019, now added as Ref. [69]) has also been added.

Referee # 3

Comment 3.1: The revision is satisfactory. I would like to recommend the acceptance of this paper.

Response 3.1: We thank the reviewer for recommending the acceptance of this manuscript.

REVIEWERS' COMMENTS

Reviewer #1 (Remarks to the Author):

Referee # 1

Comment 1.2: For instance, the differences of PPMS and TG results are notably large in 40-200 K. The temperature dependence of both is largely different.

My opinions: The explanation seems fine as the PPMS sample is relatively large and the measurement results are an average of such a "large-sized" sample. With regards to TG method, however, the signals come from a small zone, the size of which is determined by the diameter of illuminating light (0.5 mm). In this case, the measurement results are an average of "small-sized" sample. Given the discontinuity and instability (e.g., cracks) in samples particular with low-temperature measurement, these two measurements might end up with difference.

Moreover, other disparities between PPMS and TG lie in the fact that PPMS is an on-contact method. That is the heating and measuring components are in direct contact with samples, which might result in the errors caused by the surface heat radiation among heating, samples, and those components. Such errors need to be rectified, for which different methods leads to different results.

Furthermore, in TG methods of thermal conductivity calculation, the heat capacity (C_v) values adopted from literature would vary over temperature and the variation trend might differ from the real samples for measurement, for instance, in the range of 40-200 K, leading to different results from the steady-state means.

In short, I think the error up to 30% between PPMS and TG results is acceptable since such a deviation would make little difference on the conclusion.

Comment 1.3: In addition, in Fig. 4c showing grating period dependent thermal conductivities of 100 and 300 K, why the data below 300 K are nearly the same? I do not see any interpretation since the values of 300 K should be greater than those of 100 K.

My opinions: The response from the authors can be accepted. When the grating period is smaller, the corresponding thermal conductivity is smaller too. By comparison, the uncertainties of the measurement are relatively larger. In this respect, the disparities of thermal conductivities between 100 and 300 K are indistinguishable even if they exist.

Revision Report #3 for MS#NCOMMS-22-52086B

entitled:

Origin of high thermal conductivity in disentangled
ultra-high molecular weight polyethylene films:
ballistic phonons within enlarged crystals

Taeyong Kim, Stavros X. Drakopoulos, Sara Ronca, and Austin Minnich

March 28, 2022

We again thank the reviewers for their time to review the revised paper and our response to their comments. We have addressed the remaining concerns point by point and made changes to the manuscript. We believe that the manuscript is now suitable for publication in Nature Communications.

Referee # 1

Comment 1.1: The explanation seems fine as the PPMS sample is relatively large and the measurement results are an average of such a “large-sized” sample. With regards to TG method, however, the signals come from a small zone, the size of which is determined by the diameter of illuminating light (0.5 mm). In this case, the measurement results are an average of “small-sized” sample. Given the discontinuity and instability (e.g., cracks) in samples particular with low-temperature measurement, these two measurements might end up with difference.

Moreover, other disparities between PPMS and TG lie in the fact that PPMS is an on-contact method. That is the heating and measuring components are in direct contact with samples, which might result in the errors caused by the surface heat radiation among heating, samples, and those components. Such errors need to be rectified, for which different methods leads to different results.

Furthermore, in TG methods of thermal conductivity calculation, the heat capacity (C_v) values adopted from literature would vary over temperature and the variation trend might differ from the real samples for measurement, for instance, in the range of 40-200 K, leading to different results from the steady-state means.

In short, I think the error up to 30% between PPMS and TG results is acceptable since such a deviation would make little difference on the conclusion.

Response 1.1:

We thank the reviewer for providing an insightful comparison between the PPMS and the TG measurements. We agree with the reviewer's final conclusion regarding the negligible impact of the difference between PPMS and TG measurements on the conclusion.

Comment 1.2: The response from the authors can be accepted. When the grating period is smaller, the corresponding thermal conductivity is smaller too. By comparison, the uncertainties of the measurement are relatively larger. In this respect, the disparities of thermal conductivities between 100 and 300 K are indistinguishable even if they exist.

Response 1.2: We appreciate the reviewer's positive feedback of our explanation.